materials science/physical chemistry

gold quantum dots, fluorescence, grating-coupled surface plasmon, inverted organic solar cells

**Authors for correspondence:**
Kontad Ounnunkad
e-mail: kontad.ounnunkad@cmu.ac.th
Akira Baba
e-mail: ababa@eng.niigata-u.ac.jp

†Present address: Advanced Research Laboratories, Tokyo City University, 8-15-1 Todoroki, Setagaya-ku, Tokyo 158-0082, Japan.

This article has been edited by the Royal Society of Chemistry, including the commissioning, peer review process and editorial aspects up to the point of acceptance.

# The effect of gold quantum dots/grating-coupled surface plasmons in inverted organic solar cells

Kulrisa Kuntamung[1,2], Patrawadee Yaiwong[1,2], Chutiparn Lertvachirapaiboon[1], Ryousuke Ishikawa[1,†], Kazunari Shinbo[1], Keizo Kato[1], Kontad Ounnunkad[2,3,4,5] and Akira Baba[1]

[1]Graduate School of Science and Technology, Niigata University, 8050 Ikarashi-2-nocho, Nishi-ku, Niigata 950-2181, Japan
[2]Department of Chemistry, Faculty of Science, [3]Center of Excellence for Innovation in Chemistry, Faculty of Science, [4]Research Center on Chemistry for Development of Health Promoting Products from Northern Resources, and [5]Center of Excellence in Materials Science and Technology, Chiang Mai University, Chiang Mai 50200, Thailand

AB, 0000-0002-2633-8195

We studied the effect of gold quantum dots (AuQDs)/grating-coupled surface plasmon resonance (GC-SPR) in inverted organic solar cells (OSCs). AuQDs are located within a GC-SPR evanescent field in inverted OSCs, indicating an interaction between GC-SPR and AuQDs' quantum effects, subsequently giving rise to improvement in the performance of inverted OSCs. The fabricated solar cell device comprises an ITO/TiO₂/P3HT : PCBM/PEDOT : PSS : AuQD/silver grating structure. The AuQDs were loaded into a hole transport layer (PEDOT : PSS) of the inverted OSCs to increase absorption in the near-ultraviolet (UV) light region and to emit visible light into the neighbouring photoactive layer, thereby achieving light-harvesting improvement of the device. The grating structures were fabricated on P3HT:PCBM layers using a nanoimprinting technique to induce GC-SPR within the inverted OSCs. The AuQDs incorporated within the strongly enhanced GC-SPR evanescent electric field on metallic nanostructures in the inverted OSCs improved the short-circuit current and the efficiency of photovoltaic devices. In comparison with the reference OSC and OSCs with only green AuQDs or only metallic grating, the developed device indicates enhancement of up to 16% power conversion efficiency. This indicates that our light management approach allows for greater light utilization of the OSCs because of the synergistic effect of G-AuQDs and GC-SPR.

ROYAL SOCIETY OF CHEMISTRY

# 1. Introduction

The development of high-performance organic thin-film solar cells (OSCs) for harvesting and converting sunlight into electricity has increased rapidly in the past few decades. These cells serve as alternative energy sources for replacing traditional silicon solar cells, based on their potential low cost, light weight, simple preparation and good mechanical flexibility [1–5]. Among the various types of OSCs, inverted OSCs have received significant attention because of their superior device performance and better air stability compared with conventional OSCs [6,7]. In addition, their fabrication avoids the use of the acidic PEDOT : PSS on an indium tin oxide (ITO) surface and unstable low-work-function metal electrodes (typically Al or Ca) used in conventional OSC types [6,7]. For inverted structures, the mechanism that electrons employ to move to the ITO side (cathode), where holes flow to the top side and are collected on a metal layer (anode, typically Ag or Au), is proposed [8].

Reports show that introducing plasmonic, diffraction and scattering structures in solar cells can increase their device efficiency [9,10]. For example, metallic nanoparticles (NPs) introduced into OSC devices enhanced the light-harvesting ability via the scattering effect and the localized surface plasmon resonance (LSPR) effect and accordingly enhanced the synergistic effect of both [11,12]. In addition to improvements made by metallic NPs, metallic nanostructures can be used to improve the efficiency of OSCs through light scattering and propagating SPR excitation, i.e. grating-coupled SPR (GC-SPR) [13–16]. We also demonstrated OSC performance improvement using gold NPs (AuNPs) [17], silver NPs (AgNPs) [18] and metallic nanostructures/gratings [15,18–20]. These provide better light scattering and trapping and enhance the SPR effect inside the photovoltaic cells, both of which significantly contribute to increasing their light absorption [20].

Recently, quantum dots (QDs) have attracted attention as fluorescent nanomaterials for improving the efficiency of OSC devices, including cadmium selenide QDs (CdSeQDs) [21], lead sulfide QDs (PbSQDs) [22], cadmium sulfide QDs (CdSQDs) [5], graphene QDs (GQDs) [23] and gold QDs (AuQDs) [24]. AuQDs, which consist of 5–25 Au atoms (smaller than 2 nm in diameter, i.e. smaller than plasmonic AuNPs), are known to be a unique material as they do not have LSPR excitation and exhibit quantum effects that enable the UV light absorption and fluorescence emission in the visible region [25–28]. When incorporating these AuQDs into OSC devices, AuQDs displayed enhanced light-harvesting properties for the OSCs because they were able to absorb ultraviolet (UV) light and subsequently emitted fluorescent light in the visible region that overlapped with the light absorption range of the P3HT : PCBM active layer [24]. Therefore, improving device performance using AuQDs presents interesting possibilities for achieving higher device efficiencies. Moreover, by combining AuQDs and AuNPs, synergistic effects were obtained because the fluorescence from AuQDs enhances the localized plasmon effect of the AuNPs, and simultaneously the enhanced electric field on AuNPs enhances the fluorescence of AuQDs by LSPR excitation [29,30]. Furthermore, we reported that, by introducing both AuQDs and plasmonic grating structures (the GC-SPR effect) into OSCs, the multiple effect was obtained to improve their energy conversion performance via SPR excitation of metallic nanostructures, enhancing the fluorescence of AuQDs under solar illumination [31]. However, the enhancement of the fluorescence from the AuQDs by GC-SPR excitation was not observed because the AuQDs was located far from the metal grating surface in the OSCs.

Herein, we investigate an inverted OSC device by combining the application of AuQDs and the GC-SPR technique to improve the performance of both, where the AuQDs are located within a strongly enhanced GC-SPR electric field. The effect of incorporating three types of AuQDs, i.e. blue (B-AuQDs), green (G-AuQDs) and red (R-AuQDs), on cell performance was examined. The AuQDs were introduced into a poly(3,4-ethylenedioxythiophene) : poly(styrenesulfonate) layer of inverted OSCs. Two types of diffraction-grating patterns (BD-R and DVD-R) were used for the construction of metallic gratings by nanoimprinting the grating patterns onto P3HT : PCBM films. The developed architecture of our inverted OSCs was based on an $ITO/TiO_2/P3HT : PCBM/PEDOT : PSS/AuQD/Ag$ grating electrode. We found that the best light-harvesting performances of our inverted OSC devices were achieved by incorporating G-AuQDs and an Ag-DVD-R grating, which was obtained from the fluorescent effect of both GC-SPR and AuQDs.

# 2. Material and methods

## 2.1. Chemicals and materials

Three different types of AuQDs were obtained from Dai Nippon Toryo Co., Ltd. (Japan): B-AuQDs (mixed 5 and 8 Au atoms), G-AuQDs (13 Au atoms) and R-AuQDs (25 Au atoms). Poly(3,4-ethylenedioxythiophene) :

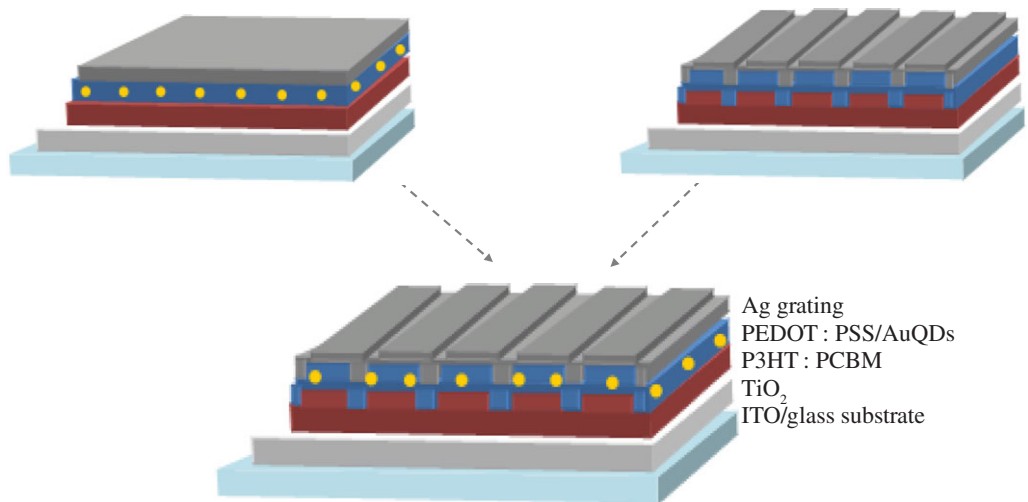

**Figure 1.** Schematic of the structures of developed devices: the AuQD device (left), grating device (right) and grating structural device incorporated with AuQDs (bottom).

poly(styrene sulfonate) (PEDOT : PSS, Clevios™ HTL Solar) was purchased from Heraeus (Germany). Poly(3-hexylthiophene-2,5-diyl) (P3HT, greater than 99%), [6,6]-phenyl $C_{61}$ butyric acid methyl ester (PCBM, greater than 99%), nitric acid ($HNO_3$, 60%), hydrochloric acid (HCl, 37%), titanium tetrabutoxide (97%), 1,2-dichlorobenzene (99%), ethanol (99.5%) and acetone (AR grade) were purchased from Sigma-Aldrich (Japan). An ITO-coated glass slide (conductivity of $10 \, \Omega \, cm^{-2}$) was purchased from Furuuchi Chemical (Japan).

## 2.2. Fabrication of inverted OSCs

The inverted OSC devices were fabricated with a structure sequence of ITO/$TiO_2$/P3HT : PCBM/PEDOT : PSS (with or without AuQDs)/Ag (with or without grating structures) as described in figure 1. To obtain DVD-R (grating pitch, $\Lambda = 740$ nm) and BD-R ($\Lambda = 320$ nm) master templates, DVD-R and BD-R were cut into small pieces. After removing the dye coating, the DVD-R and BD-R pieces were sequentially washed with detergent, tap water and deionized water. Liquid polydimethylsiloxane (PDMS) was degassed and poured onto cut-DVD-R and BD-R. Patterns of PDMS moulds with negative DVD-R and BD-R were created after curing for 1 h at 79°C.

In the fabrication process, the device was constructed on a clean ITO-coated glass substrate (surface area 1.0 $cm^2$). The ITO glass substrate was treated with UV/ozone for 45 min to remove organic residues and improve surface wettability. Titanium tetrabutoxide (250 µl), as a Ti precursor, was stirred for 1 h in ethanol (5.0 ml). Then, the mixture (150 µl) of water and $HNO_3$ (1 : 5) was added to the above solution. The reaction mixture was carried out at room temperature with stirring for 3 h. The sol was transformed into a gel after ageing the sol for 24 h. The obtained titanium dioxide ($TiO_2$) solution was first deposited on an ITO/glass substrate by a spin-coating technique at a spin rate of 2000 rpm for 60 s and preheated at a temperature of 100°C for 10 min to remove the solvent. The $TiO_2$ thin film was thermally annealed at a temperature of 400°C for 30 min on a hot plate. This procedure was repeated twice to achieve the desired film thickness and smoothness. For active layer, a P3HT : PCBM blend dissolved in 1,2-dichlorobenzene with a mass ratio of 1 : 0.8 was spin-coated on top of the $TiO_2$ film at a spin rate of 1500 rpm for 20 s and then at a spin rate of 2000 rpm 40 s and dried at 100°C for 1 h to construct the photoactive layer of the device. Subsequently, the PDMS pattern (DVD-R or BD-R) was placed on the surface of P3HT : PCBM film by using a nanoimprinting technique. In this process, unwanted bubbles trapped at the interface between the PDMS mould and P3HT : PCBM film were removed by vacuuming before annealing at 100°C for 1 h. After cooling down, the PDMS mould was peeled from the device. To prepare the PEDOT : PSS : AuQD solution, the AuQD solution (B-AuQDs, G-AuQDs or R-AuQDs) with an optimized concentration was mixed with the PEDOT : PSS solution (ratio of 1 : 6 v/v) by sonication for 1 h. The mixture of the PEDOT : PSS : AuQD solution was then spin-coated on a grating-patterned P3HT : PCBM layer at a spin rate of 1500 rpm for 90 s and then baked at a temperature of 120°C for 10 min to evaporate the solvent. Finally, a 150 nm Ag electrode was thermally evaporated onto the PEDOT : PSS : AuQD layer. All of the fabricated devices were

annealed in a vacuum chamber at a temperature of 150°C for 45 min to obtain the better contact between each film before further characterization. The schematic diagram of the fabrication process for our developed inverted OSCs is shown in electronic supplementary material, figure S1.

## 2.3. Device characterization

The photovoltaic properties and impedance spectra of the inverted OSCs were measured using an electrometer (B2901, Agilent) and a potentiostat (PARSTAT 4000, Princeton Applied Research), respectively. Measurements were performed under illumination provided by a solar simulator (HAL-C100, Asahi Spectra USA Inc.) at a light intensity of 75 mW cm$^{-2}$. The optical properties of the fabricated devices were characterized using a homemade reflectometer on a $\theta$ to 2$\theta$ goniometer with a tungsten-halogen light source (HL-2000, Ocean Optics, Inc.). The UV–visible absorption spectra of the AuQDs, PEDOT : PSS and PEDOT : PSS : AuQD solutions were investigated using a UV–vis spectrometer (V-650, Jasco). The morphologies of the fabricated films were monitored using an atomic force microscope (AFM, SPM-9600, Shimadzu, Japan).

# 3. Results and discussion

## 3.1. The effect of three types of AuQDs on flat inverted OSCs

In the present study, we investigated three types of AuQDs, including B-AuQDs, G-AuQDs and R-AuQDs, in a PEDOT : PSS and compared the results with that of reference-inverted OSCs. The use of AuQDs was expected to increase light-trapping within devices by converting UV light to visible light. The different sizes of AuQDs offered the emission of a variety of fluorescent colours, which affected the performance of devices [17,29]. First, we examined the effect of three types of AuQDs in the inverted OSCs without a grating structure. The concentration of AuQDs in the PEDOT : PSS layers varied from 0.14 to 1.00 µM (figure 2). The highest power conversion efficiency (PCE) values for all cases were observed at 0.43 µM. Efficiency gradually increased as the concentration of AuQDs increased up to 0.43 µM before decreasing. Since a higher concentration of AuQDs caused aggregation, the result in this instance may be decreased efficiency. The significant aggregation of AuQDs was observed for B-AuQDs, and this was likely the cause of the decreased device efficiency compared with that of G-AuQD-based inverted OSCs. This result was similar to that presented in a previous report [29]. Moreover, G-AuQDs provided fluorescence that could be harvested by the absorption of the photoactive layer, accordingly providing the best PCE value. As the G-AuQDs at a concentration of 0.43 µM showed the best performance for the device, G-AuQDs (0.43 µM) were used in subsequent experiments. Figure 3 shows fluorescence spectra of AuQDs, AuQDs/PEDOT : PSS and PEDOT : PSS aqueous solutions. The G-AuQDs absorbed light in a wavelength region lower than 450 nm and then emitted fluorescence at a wavelength of roughly 510 nm [24,29], and this provided a good match for the absorption region of the P3HT : PCBM active layer. Generally, the P3HT : PCBM exhibited weak photoelectric conversion in the UV region. Thus, G-AuQDs had a considerable impact as an additional photosensitizer for harvesting UV light from sunlight and converting it to visible light within the active layer. In this way, improved inverted OSC performance was achieved. We also studied the effect of incorporating G-AuQDs in the PEDOT : PSS layers on their fluorescence intensity. In this experiment, G-AuQDs were mixed into the PEDOT : PSS aqueous solution because of their solubility in water. The results showed that G-AuQDs retained similar absorption and fluorescence characteristics. With PEDOT : PSS in the solution, fluorescence intensity was lowered (figure 3). This was the result of masking and fluorescence-absorption by PEDOT : PSS. However, it should be noted that G-AuQDs surrounded by PEDOT : PSS exhibited the ability to provide fluorescent light to other components in the device. It is plausible that the remaining fluorescence from G-AuQDs loaded into the PEDOT : PSS films could be harvested by the photoactive layer within the inverted OSC device, thus enhancing PCE. This shows good agreement with the performances of invested OSCs listed in table 1 and with current density–voltage ($J$–$V$) properties noted in electronic supplementary material, figure S2. The $J_{sc}$ values were 7.29 mA cm$^{-2}$, 6.90 mA cm$^{-2}$ and 7.18 mA cm$^{-2}$ for the G-, R- and B-AuQD-based devices, respectively. An increase in the PCE (compared with those of the reference cell) was observed, with increased values of 3.60% (10.77% relative improvement), 3.42% (5.23% relative improvement) and 3.41% (4.92% relative improvement) for the G-, R- and B-AuQD-loaded OSCs, respectively. No significant change in $V_{oc}$ was observed for any cells. These enhancements

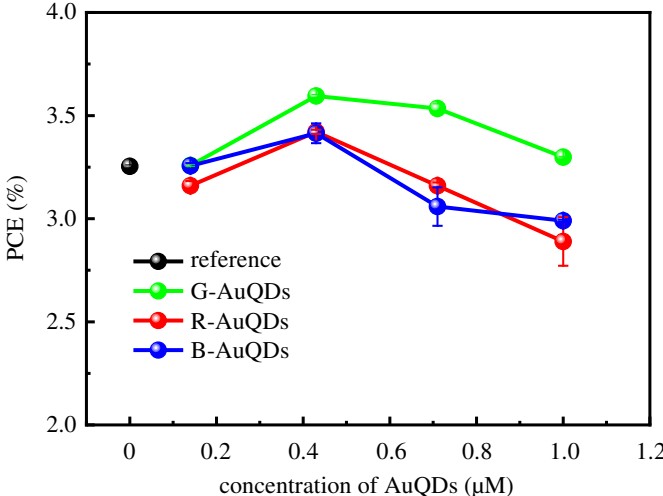

**Figure 2.** The efficiency of the AuQD-loaded inverted OSCs without a grating structure as a function of AuQD concentration, compared with that of a reference-inverted OSC.

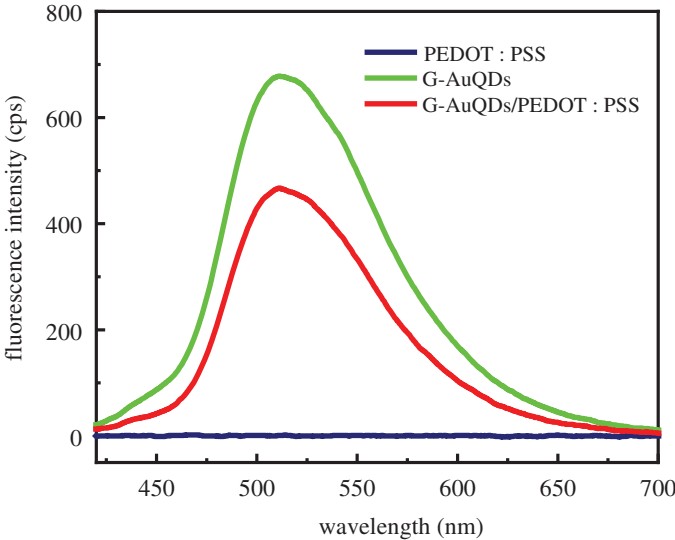

**Figure 3.** Fluorescence spectra of AuQDs, AuQDs/PEDOT : PSS and PEDOT : PSS aqueous solutions ($\lambda_{ex} = 405$ nm).

**Table 1.** Photovoltaic parameters of the inverted flat OSCs incorporated with three types of AuQDs.

| devices | $J_{sc}$ (mA cm$^{-2}$) | $V_{OC}$ (V) | FF (%) | PCE (%) | relative improvement (%) |
|---|---|---|---|---|---|
| reference | 6.88 | 0.64 | 0.55 | 3.25 | — |
| G-AuQDs | 7.29 | 0.65 | 0.57 | 3.60 | 10.77 |
| R-AuQDs | 6.90 | 0.64 | 0.58 | 3.42 | 5.23 |
| B-AuQDs | 7.18 | 0.64 | 0.56 | 3.41 | 4.92 |

occurred because AuQDs acted as effective photosensitizers for absorbing light in the UV region and the photoactive layer harvested the emitted visible light, contributing to the generation of more photo-generated carriers [25,30]. It is noted that the effect of AuQDs on the enhancement of the developed inverted OSCs (10.77% relative improvement for G-AuQDs) is more significant than that of previously studied conventional standard-OSCs (8.26% relative improvement for G-AuQDs) [29,31]. One possibility for the higher effect of AuQDs in the inverted OSCs is that the fluorescence light reflected

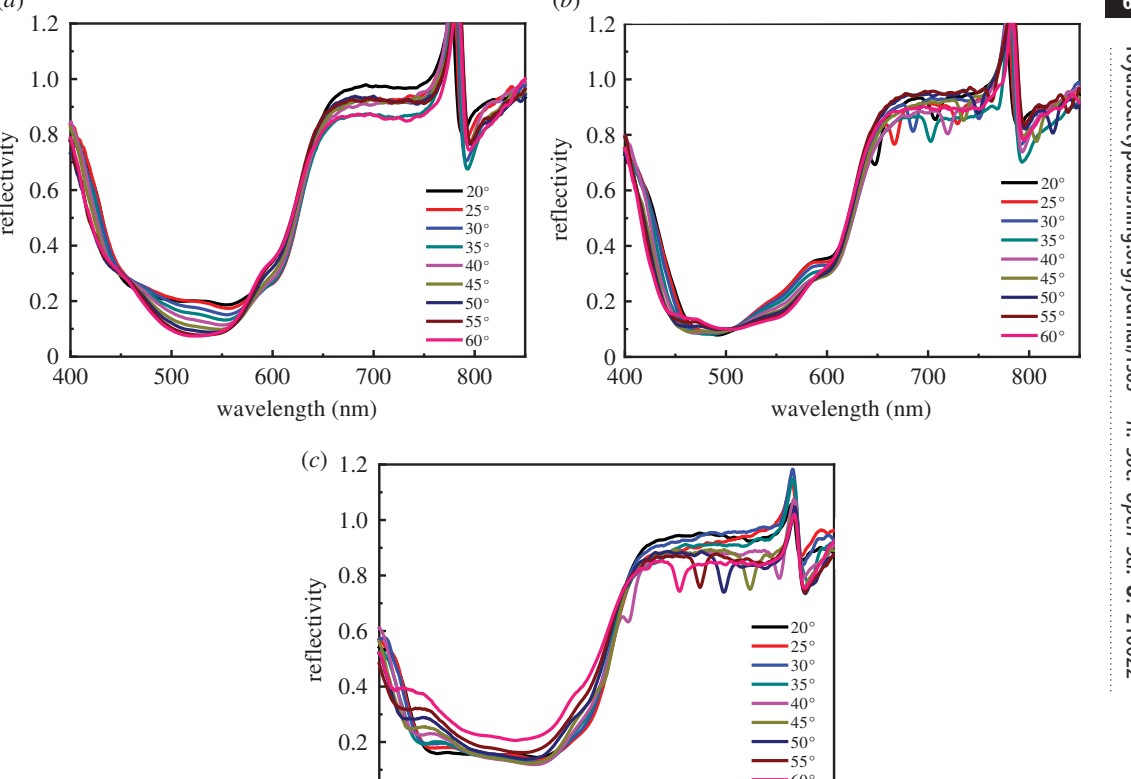

**Figure 4.** (*a*) Reflectivity curves of silver-coated flat surface, (*b*) the G-AuQDs/BD-R grating and (*c*) the G-AuQDs/DVD-R grating of inverted OSCs under p-pol illumination.

at the Ag electrode is additionally attributed to the enhancement in the photocarrier generation in the P3HT : PCBM layer.

## 3.2. Optical properties and surface morphology of fabricated inverted OSCs

We examined the optical properties of fabricated devices to confirm GC-SPR excitation on the Ag grating electrodes, as shown in figure 4. A p-polarized (p-pol) white light was irradiated on the OSCs from the ITO side as a function of wavelength at fixed incident angles from 20° to 80°. For a flat Ag electrode-based inverted OSC (figure 4*a*), no SPR dip peak was observed over any incident light angles present in the reflectivity spectra. The broad low reflectivity at a wavelength range from 400 to 650 nm indicated light absorption of the inverted OSC, primarily in the P3HT : PCBM photoactive layer. A lowered reflectivity at higher incident angle might be due to multiple reflection in the device. In the case of devices with BD-R and DVD-R Ag grating electrodes (as shown in figure 4*b,c*), the wavelength region of broad low reflectivity was further extended, particularly for the device with a DVD-R grating electrode. This was due to light scattering or weak SPR excitations in this region, leading to more absorption in the P3HT : PCBM photoactive layer. In particular, for the device with a DVD-R grating electrode, a dip at roughly 400–460 nm was observed. The dip wavelength depended on the incident angle, indicating GC-SPR excitation on the Ag grating. In addition, a dip originated from GC-SPR or waveguide mode [19] in the range of 650–800 nm was observed in devices with BD-R and DVD-R Ag grating electrodes, respectively. This result confirms that GC-SPR on pattern-based Ag grating electrodes could be excited in this configuration. The DVD-R grating pattern provided deeper SPR dips than the BD-R grating because each grating pitch height of the former pattern was higher than that of the latter (see electronic supplementary material, figures S3 and S4). Electronic supplementary material, figures S3 and S4 indicate that the grating patterns were successfully transferred from the master templates to the P3HT : PCBM layer, and the grating heights of nanoimprinted BD-R and DVD-R grating structures on the P3HT : PCBM were roughly 10 nm and 31 nm, respectively.

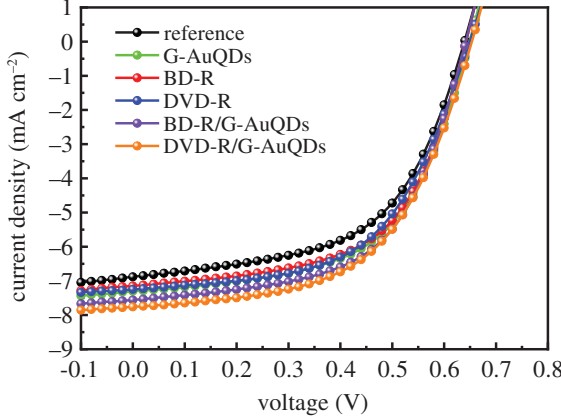

**Figure 5.** The *J–V* characteristics of the inverted OSCs based on the G-AuQDs and/or plasmonic grating structures compared with the reference cell.

**Table 2.** Comparison of the photovoltaic parameters of fabricated inverted OSCs with different nano-architectures.

| devices | $J_{sc}$ (mA cm$^{-2}$) | $V_{oc}$ (V) | FF (%) | PCE (%) | relative improvement (%) |
|---|---|---|---|---|---|
| reference | 6.88 | 0.63 | 0.55 | 3.25 | — |
| G-AuQDs | 7.29 | 0.65 | 0.57 | 3.60 | 10.77 |
| BD-R | 7.14 | 0.64 | 0.57 | 3.48 | 7.08 |
| DVD-R | 7.24 | 0.65 | 0.56 | 3.50 | 7.69 |
| BD-R/G-AuQDs | 7.57 | 0.64 | 0.58 | 3.70 | 13.85 |
| DVD-R/G-AuQDs | 7.75 | 0.65 | 0.56 | 3.77 | 16.00 |

## 3.3. Photovoltaic performance of AuQDs/grating-structured inverted OSCs

To understand the effect of grating structures on enhancing inverted OSCs, we investigated two grating nanopatterns involving BD-R and DVD-R grating structures for the construction of the devices to be compared, using a flat structure as a reference. Figure 5 indicates the *J–V* properties, and table 2 summarizes the photovoltaic parameters. The results indicated that a higher photocurrent was obtained from the BD-R and DVD-R devices compared with the reference cell. The DVD-R and BD-R grating-based devices showed $J_{sc}$ values of 7.24 and 7.14 mA cm$^{-2}$, and PCE values of 3.50% (7.69% relative improvement) and 3.48% (7.08% relative improvement), respectively, with slight changes in the $V_{oc}$ and the fill factor (FF). The enhancements can be explained by the effect of the GC-SPR and light scattering provided by the grating structures, resulting in the improvement of the inverted OSCs' performance [19]. In comparing the efficiencies of the DVD-R and BD-R devices, we can observe that the DVD-R device provided better enhancement than the BD-R device because of stronger GC-SPR excitation and scattering. However, it is also noted that the grating effect in the developed inverted OSCs (7.08% relative improvement for BD-R) is less than that in previously studied conventional standard-OSCs (10.40% relative improvement for BD-R) [31]. This is reasonable because the P3HT : PCBM in the inverted OSCs is located in the weaker GC-SPR evanescent field due to the presence of PEDOT : PSS layer between the P3HT : PCBM and Ag grating films. Moreover, the grating height in the inverted OSCs (electronic supplementary material, figure S4) is lower than that in the previously studied standard-OSCs due to the presence of TiO$_2$ layer, resulting in the weaker SPR excitation as shown in figure 4.

After studying the individual effects of incorporating AuQDs and grating structures in the inverted OSCs, we investigated the possible multiple enhancement effect of putting AuQDs within the GC-SPR evanescent field on Ag grating structures in the inverted OSCs. In this study, G-AuQDs were chosen to fabricate the device with an Ag top electrode grating structure (BD-R or DVD-R pattern) because they exhibited the best efficiency compared with those of R- and B-AuQDs. Under the same

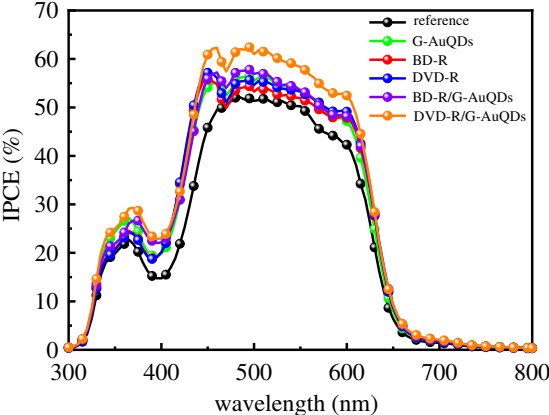

**Figure 6.** The IPCE spectra of inverted OSCs based on the G-AuQDs and/or plasmonic grating structures compared with the reference cell.

conditions as above, G-AuQDs were added to the PEDOT : PSS layer for UV light harvesting, and a grating structure was created on the surface of a photoactive layer and an Ag top electrode. This was achieved by vacuum evaporation for photon trapping and by generating GC-SPR excitation [19]. The $J$–$V$ characteristics of different G-AuQDs/grating structures of fabricated solar cells are presented in figure 5, and the photovoltaic parameters of the devices are given in table 2. After incorporating G-AuQDs and/or a grating structure (DVD-R or BD-R) in the devices, the $V_{oc}$ and FF were found to be similar, the $J_{sc}$ had increased from 7.14 (BD-R) and 7.24 (DVD-R) mA cm$^{-2}$ to 7.57 (BD-R/G-AuQDs) and 7.75 (DVD-R/G-AuQDs) mA cm$^{-2}$, and PCE relative improvement had increased from 7.08% (BD-R) and 7.69% (DVD-R) to 13.85% (BD-R/G-AuQDs) and 16.0%(DVD-R/G-AuQDs), respectively, compared with those of the reference cell. Interestingly, the G-AuQDs/grating structure provided improved photovoltaic performances compared with only G-AuQDs or having only the grating structure in the device. This was because the former presented synergistic benefits in the light management of the developed devices. By comparing the efficiencies of devices with the G-AuQD/ grating structure and those with only the grating structure (without G-AuQD), the G-AuQD/grating structure was found to provide significantly better PCE and $J_{sc}$ values. This evidence suggests that synergistic enhancement can be obtained from each G-AuQD/grating system. The enhancement of the conversion efficiency of the inverted OSC, based on G-AuQD/BD-R grating (PCE = 3.70%) and compared with that of the device with a BD-R grating (PCE = 3.48%), was found to be 6.3%. The enhancement of the conversion efficiency of the device employing a G-AuQD/DVD-R grating system (PCE = 3.77%) was 7.7% compared with the OSC-based device with only a DVD-R grating (PCE = 3.50%). For binary systems, improvements in the PCE and $J_{sc}$ values of the inverted OSC by cell fabrication, combined with G-AuQDs and DVD-R grating architecture, were observed, and the results supported that only the DVD-R grating structure offered an increase in both parameters compared with the device using only a BD-R grating pattern. Therefore, the observation above indicates that the best synergistic effect was obtained in this paper by the G-AuQD/DVD-R grating system. It is noted that although obvious synergistic effect was obtained, the PCE relative improvement of BD-R gating with G-AuQDs in the developed inverted OSCS is slightly smaller than that in previously studied standard-OSCs with BDR-R grating structure and G-AuQDs [31]. As mentioned above, this might be due to weak GC-SPR excitation in the inverted OSCs due to the lower grating height.

To study the effect of the presence of both G-AuQDs and a grating structure in the inverted OSCs, we measured their incident photon-to-current efficiency (IPCE) as shown in figure 6. Compared with the reference cell, the solar cells fabricated with individual systems (only G-AuQDs or only a grating structure) exhibited higher IPCE values over a broader wavelength range, which is similar to our previous study using conventional OSCs [20,24]. Furthermore, incorporating both G-AuQDs and a grating structure into devices achieved higher IPCE values than those for individual systems. The IPCE spectrum of the OSC based on the G-AuQDs/DVD-R grating system showed the highest IPCE improvement. In the case of inverted OSCs with added G-AuQDs, the IPCE was largely enhanced in the range of roughly 380–450 nm, which was likely the result of light utilization by G-AuQDs in the device. In the case of inverted OSCs with gratings, the enhancement in the broader wavelength range can be considered as the effect of GC-SPR and light scattering, as was observed in the SPR reflectivity results.

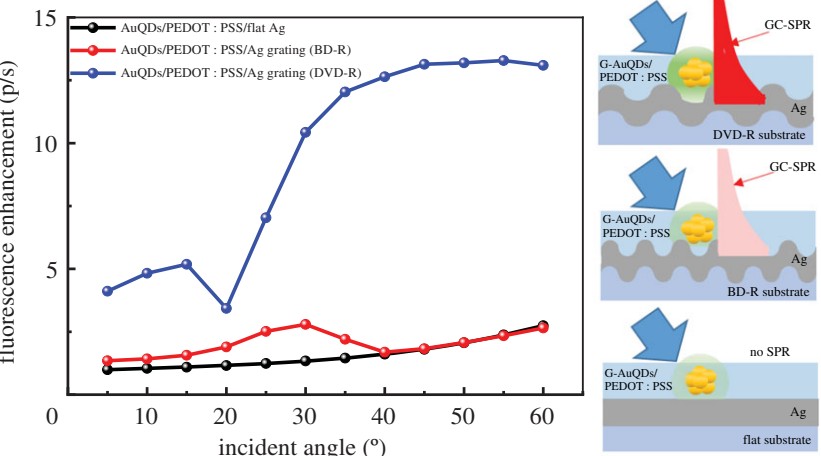

**Figure 7.** The fluorescence enhancement (p/s) of the G-AuQDs/PEDOT : PSS film deposited on flat Ag and G-AuQDs/PEDOT : PSS films layered on Ag-coated BD-R and DVD-R gratings. The fluorescence of G-AuQDs at 510 nm was recorded under illumination at a wavelength of 300–500 nm.

## 3.4. Fluorescence and SPR reflectivity properties of the G-AuQDs/PEDOT : PSS film on an Ag grating

To further study the effect of AuQDs and GC-SPR, we examined enhancement in the fluorescence emission of G-AuQDs within a GC-SPR evanescent field on Ag gratings. When the GC-SPR was excited, a strongly enhanced electric field was generated near the Ag surface. The enhanced electric field from the GC-SPR enhanced the fluorescence if the fluorophore material was located within the GC-SPR evanescent field [32]. Figure 7 shows the fluorescence enhancement (p/s) at 510 nm of the G-AuQDs/PEDOT : PSS film deposited on flat Ag and G-AuQDs/PEDOT : PSS films coated on Ag-coated BD-R and DVD-R gratings under illumination at an excitation wavelength of 300–500 nm. The p/s was obtained from fluorescence intensity with p-pol light illumination divided by fluorescence intensity with s-pol light illumination. Since GC-SPR can be excited only by the illumination of p-pol light, fluorescence enhancement by GC-SPR in this plot can be evaluated. To check GC-SPR excitation, the SPR reflectivity curves were also measured for the same sample used in figure 7 and the SP dispersion properties are also plotted as shown in figure 8. As shown in figure 8, GC-SPR with BD-R grating corresponded to $m = -SP^{+1}$ mode, while the GC-SPR with DVD-R corresponded to $m = +SP^{-1}$ and $-SP^{+2}$ modes. In the case of the BD-R grating, the p/s was increased 2.5 times at an incident light angle of 30°. Interestingly, the GC-SPR dip wavelength at 30° was at roughly 520 nm (figure 8a), where the G-AuQDs exhibited strong fluorescence (figure 3). The enhanced fluorescence may have been due to the emission of GC-SPR excited by the G-AuQDs fluorescence, in addition to the G-AuQDs fluorescence itself. The fluorescent light from the G-AuQDs induced GC-SPR on the Ag surface, and the latter was able to emit from the surface because of the roughness of the Ag surface [32]. At above 40°, p/s on the BD-R grating was almost completely the same as that on the flat surface. This was considered reasonable because the GC-SPR excitation wavelength at above 40° (figure 8a) was far removed from the AuQDs of both G-AuQDs' absorption wavelength and fluorescence wavelength. In the case of the DVD-R Ag grating, significant p/s was observed. It was observed that the p/s reached a maximum value of 13-fold at an incident light angle of 40° and then remained constant. As shown in figure 8b, the reflectivity curves indicated broad low reflectivity in the wide wavelength region, which indicated light scattering and GC-SPR excitation on the surface, leading to the enhancement of G-AuQDs' fluorescence. Since reflectivity indicated broad and complicated properties, it was not obvious what the relation was between the GC-SPR wavelength and the G-AuQDs' absorption and fluorescence wavelength. However, it is noted that the $m = +SP^{-1}$ mode in GC-SPR shifted from 560 nm at 20° to 515 nm at 40° and to 483 nm at 50°, indicating that the GC-SPR wavelength above 30° matched the wavelength region of G-AuQD fluorescence and absorption. Hence, one reason for the significant enhancement of fluorescence intensity of G-AuQDs had been the interaction between the GC-SPR and the G-AuQDs alongside the scattering effect. Since the G-AuQDs were located within the GC-SPR evanescent field (shorter than 500 nm), the enhanced GC-SPR electric field improved the fluorescence intensity at roughly 510–

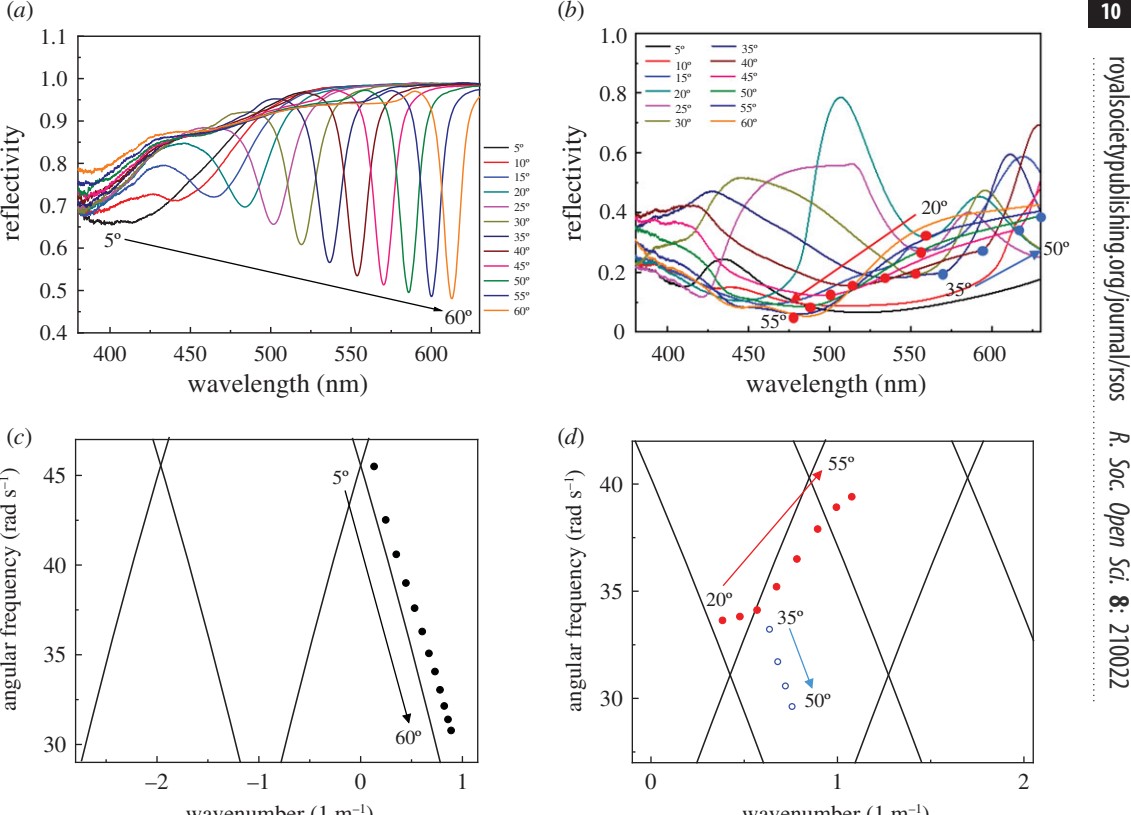

**Figure 8.** The SPR reflectivity curves of (*a*) the AuQDs/PEDOT : PSS film on the BD-R-patterned Ag grating and (*b*) the AuQDs/PEDOT : PSS film on an Ag-deposited DVD-R grating substrate. Theoretical (solid lines) and experimental (dots) SP dispersion relations of (*c*) the AuQDs/PEDOT : PSS film on the BD-R-patterned Ag grating and (*d*) the AuQDs/PEDOT : PSS film on an Ag-deposited DVD-R grating substrate.

540 nm. In addition, the fluorescence from G-AuQDs at roughly 510–540 nm induced additional GC-SPR excitation at the same wavelength, which may have been emitted from the Ag grating surface on account of the surface's roughness. These results showed good agreement with photovoltaic results (figures 5 and 6) in which the best improvement in solar cells' performance was observed with the dual G-AuQDs/DVD-R-patterned Ag grating top electrode system.

## 3.5. Impedance spectroscopy of AuQDs/grating inverted OSCs

To study the internal resistances and carrier transport of the devices, electrochemical impedance spectroscopy spectra were measured. Figure 9 shows Nyquist plots of the inverted OSCs based on the G-AuQDs and/or grating structures compared with that of the reference cell. Electronic supplementary material, table S1 summarizes the resistance ($R_s$ and $R_{ct}$) values for all developed inverted OSCs. It was found that the $R_s$ values of the AuQD-loaded devices were slightly smaller than those of the reference device and grating-only devices, indicating that the resistance of fabricated devices decreased by adding AuQDs into the PEDOT : PSS layer. The $R_{ct}$ value corresponded to the diameter of the semicircle at a low-frequency region and represented the bulk resistance of the $TiO_2$/P3HT : PCBM interface [33]. The reason for the lowered $R_s$ and $R_{ct}$ values might be due to the enhanced hole/electron injection at the PEDOT : PSS/P3HT : PCBM interface. In addition to the photocarrier generation in P3HT : PCBM via fluorescence of AuQDs, excited electrons within AuQDs at the interface of PEDOT : PSS-AuQDs/P3HT : PCBM might be injected into the conduction band of the adjacent PCBM and, at the same time, holes might be injected into the valence band of PEDOT : PSS, which could also contribute to the increased short-circuit current [24]. The $R_{ct}$ values of individual G-AuQDs' inverted OSCs and grating-structured inverted OSCs were similar and decreased compared with the reference OSC, thus indicating the enhancement of the charge transfer properties [34]. After incorporating G-AuQDs and the grating structure, the $R_s$ values of G-AuQD/Ag grating devices were slightly lower than those of the G-AuQD-only device.

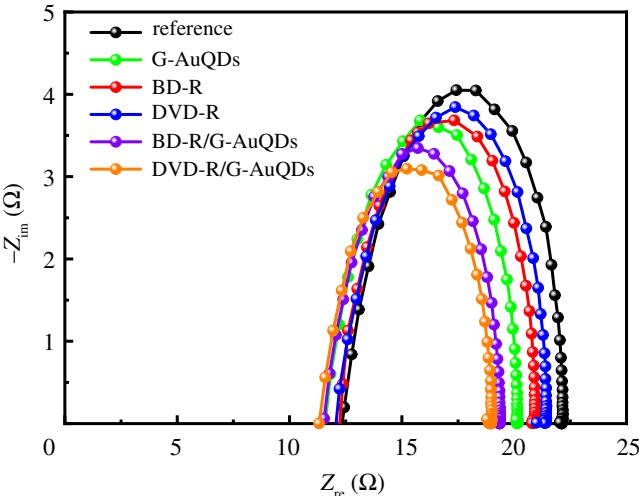

**Figure 9.** Nyquist plots of the inverted OSCs based on the G-AuQDs and/or grating structures compared with those of the reference cell.

Moreover, compared with those of individual systems and those in hybrid systems, the decreased $R_{ct}$ values of the fabricated G-AuQDs and grating-structured OSCs were observed. The reference's inverted OSC and the G-AuQD, BD-R Ag grating and DVD-R Ag grating with inverted OSCs exhibited $R_{ct}$ values of 9.9 Ω, 8.5 Ω, 8.6 Ω and 8.8 Ω, respectively, and the G-AuQD/BD-R Ag grating and G-AuQD/DVD-R Ag grating with inverted OSCs showed low $R_{ct}$ values of 7.9 Ω and 7.7 Ω, respectively. This indicated that introducing G-AuQDs into the PEDOT : PSS layer and a grating structure onto the photoactive layer with an Ag grating top electrode could enhance the charge transfer properties in devices, as indicated by the lowest $R_{ct}$. This result showed good agreement with the photovoltaic performances of figures 5 and 6. Furthermore, for all developed OSCs, the frequency peaks of Bode phase plots as they related to electron lifetime [29,35] are plotted in electronic supplementary material, figure S5, and the corresponding calculated average electron lifetime ($\tau_{avg}$) values are listed in electronic supplementary material, table S1. An insignificant difference in $\tau_{avg}$ values was observed for all devices. This result indicated that the use of AuQDs or a grating-imprinted Ag electrode, or both together, may not affect a device's electron lifetime. Therefore, this may imply that the improvement of the inverted OSC performance primarily originated from enhanced photocarrier generation, brought on by the dual effect of AuQDs and GC-SPR on the Ag grating, rather than from the extension of electron lifetime. Therefore, we conclude that for the inverted OSC based on the dual G-AuQDs/Ag grating top electrode, the best photocarrier generation was obtained as a result of the additional synergistic effect of G-AuQDs and Ag grating, which was established by observing the behaviour of GC-SPR-enhanced fluorescence emission.

## 4. Conclusion

We investigated the effect of G-AuQDs within the GC-SPR evanescent field in inverted OSCs. Although the GC-SPR excitation in the inverted OSCs is relatively weaker than that in the previously studied conventional standard-OSCs, the dual system can improve the light-harvesting performances of inverted OSCs. The inverted OSCs, based on the dual G-AuQDs/DVD-R nano-patterned Ag back electrode system, exhibited the best improvement in terms of solar cell performance. The synergistic effect of both components was observed, and a significantly enhanced fluorescence intensity for G-AuQDs within the GC-SPR evanescent field on the DVD-R-nanoimprinted Ag grating was detected. The enhanced PCE and $J_{sc}$ values of the developed inverted OSCs with the dual system were considered to have been achieved from additional light harvesting of G-AuQDs in UV and near-UV regions (consumable fluorescence for the active layer), scattering inside the cell by a DVD-R-imprinted Ag grating top electrode, the GC-SPR effect, the GC-SPR-enhanced fluorescence effect and AuQD-fluorescence-induced additional GC-SPR emission. A synergistic effect was obtained when the GC-SPR excitation wavelength on the DVD-R Ag grating electrode closely matched the fluorescence and absorption wavelength range of the G-AuQDs, thus resulting in enhanced fluorescence. Our strategy was thus identified as a promising approach for enhancing the performance of inverted OSCs, and it has the potential to develop other types of optoelectronic devices.

Data accessibility. Data are available at the Dryad Digital Repository: https://doi.org/10.5061/dryad.vdncjsxt5 [36].

Authors' contributions. Material preparation and data collection were performed by K.Ku. and P.Y. Data analysis was performed by K.Ku. and C.L., A.B. designed the study, coordinated the study and helped draft the manuscript. R.I., K.S., K.Ka. and K.O. co-supervised the project and made a valid contribution to the manuscript. The first draft of the manuscript was written by K.Ku. and all authors commented on previous versions of the manuscript. All authors read and approved the final manuscript.

Competing interests. We declare we have no competing interests.

Funding. This work was supported by Japan Society for the Promotion of Science (JSPS) KAKENHI grant nos JP20H02601 and JP20K21140. This work was also supported by the Program Management Unit for Human Resources & Institutional Development, Research and Invitation, NXPO (Grant Number: B16F640001).

Acknowledgements. K.Ku. thanks an award of the Science Achievement Scholarship of Thailand (SAST). This work was partially supported by Chiang Mai University. K.O., K.Ku. and P.Y. gratefully acknowledge the Graduate School (Chiang Mai University), Research Center on Chemistry for Development of Health Promoting Products from Northern Resources, Center of Excellence for Innovation in Chemistry (PERCH-CIC), Center of Excellence in Materials Science and Technology and Department of Chemistry, Faculty of Science, Chiang Mai University. This research was supported by the Program Management Unit for Human Resources & Institutional Development, Research and Invitation, NXPO [Frontier Global Partnership for Strengthening Cutting-edge Technology and Innovations in Materials Science].

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
