## [Peer Review File · Royal Society Open Science]

Review History

RSOS-210022.R0 (Original submission)

Review form: Reviewer 1

Is the manuscript scientifically sound in its present form?

Yes

Are the interpretations and conclusions justified by the results?

Yes

Is the language acceptable?

Yes

Do you have any ethical concerns with this paper?

No

Have you any concerns about statistical analyses in this paper?

No

Recommendation?

Accept with minor revision (please list in comments)

Comments to the Author(s)

In this manuscript, the authors have loaded the gold quantum dots (AuQDs) into PEDOT:PSS solution to study the effect on inverted organic solar cells (OSCs), which is assisted with the synergistic effect of grating-coupled surface plasmon resonance (GC-SPR). The authors have provided another possible method to enhance the light-harvesting properties for the OSCs. The results presented in this manuscript are somehow interesting, and it can be published in Royal Society Open Science after some revisions.

1. The conventional device structure is more widely used nowadays for its simple fabrication process and higher performance. However, in this manuscript, the authors only mentioned the effect on the inverted OSCs. To show the universality of this light-harvesting enhancement approach, I would recommend to show some data about the effect on OSCs with conventional structure.
2. Again, to verify its universality, it's better to add at least one more material system, whose photovoltaic performance is improved through this synergistic effect of AuQDs and GC-SPR.
3. In Figure 3, it shows the fluorescence strength of G-AuQDs is enhanced after blending with PEDOT:PSS. It would be better if the authors can explain in detail about its work mechanism.
4. In part 3.2, the authors mentioned that all the fabricated devices are annealed at 150 °C for 45 min. The purpose needs to be presented.

Review form: Reviewer 2

Is the manuscript scientifically sound in its present form?

Yes

Are the interpretations and conclusions justified by the results?

Yes

Is the language acceptable?

Yes

Do you have any ethical concerns with this paper?

No

Have you any concerns about statistical analyses in this paper?

No

Recommendation?

Major revision is needed (please make suggestions in comments)

Comments to the Author(s)

Kuntamung et al. used gold quantum dots and grating to enhance the light-harvesting of organic solar cells. By incorporating the down-conversion property of gold quantum dots and surface plasmon resonance of gratings, the authors obtained a 16% increase in power conversion efficiency. The obtained simulation results have a certain significance in light manipulation in photovoltaic devices. This article can be considered publishing on Royal Society Open Science after addressing the following questions:

1. The authors ascribed the device performance enhancement to “AuQDs acted as effective photosensitizers for absorbing light in the UV region and the photoactive layer harvested the emitted visible light”. Why the introduction of gold quantum dots results in the IPCE enhancement in the whole spectrum?
2. The down-conversion efficiency of gold quantum dots is unclear. How many ultraviolet photons absorbed by gold quantum dots can be converted to emitted photons?
3. Why the gold quantum dots boost the hole transporting in PEDOT:PSS rather than trapping the carriers?
4. According to Figure 4, the reflectivity of flat device is below 0.1 around 520 nm, while the G-AuQDs/DVD-R grating-based device has a value around 0.15 that indicating poorer light harvesting. Why the increased reflectivity leads to the increase in IPCE?

Decision letter (RSOS-210022.R0)

Dear Professor Baba:

Title: The Effect of Gold Quantum Dots/Grating-coupled Surface Plasmons in Inverted Organic Solar Cells

Manuscript ID: RSOS-210022

The editor assigned to your manuscript has now received comments from reviewers. We would like you to revise your paper in accordance with the referee and Subject Editor suggestions which can be found below (not including confidential reports to the Editor). Please note this decision does not guarantee eventual acceptance.

Please submit your revised paper before 27-Feb-2021. Please note that the revision deadline will expire at 00.00am on this date. If we do not hear from you within this time then it will be assumed that the paper has been withdrawn. In exceptional circumstances, extensions may be possible if agreed with the Editorial Office in advance. We do not allow multiple rounds of revision so we urge you to make every effort to fully address all of the comments at this stage. If deemed necessary by the Editors, your manuscript will be sent back to one or more of the original reviewers for assessment. If the original reviewers are not available we may invite new reviewers.

On behalf of the Subject Editor Professor Anthony Stace and the Associate Editor Professor Chaohua Cui.

RSC Associate Editor:
Comments to the Author:
(There are no comments.)

RSC Associate Editor:
Comments to the Author:
(There are no comments.)

Reviewers' Comments to Author:
Reviewer: 1

Comments to the Author(s)

In this manuscript, the authors have loaded the gold quantum dots (AuQDs) into PEDOT:PSS solution to study the effect on inverted organic solar cells (OSCs), which is assisted with the synergistic effect of grating-coupled surface plasmon resonance (GC-SPR). The authors have provided another possible method to enhance the light-harvesting properties for the OSCs. The results presented in this manuscript are somehow interesting, and it can be published in Royal Society Open Science after some revisions.

1. The conventional device structure is more widely used nowadays for its simple fabrication process and higher performance. However, in this manuscript, the authors only mentioned the effect on the inverted OSCs. To show the universality of this light-harvesting enhancement approach, I would recommend to show some data about the effect on OSCs with conventional structure.
2. Again, to verify its universality, it's better to add at least one more material system, whose photovoltaic performance is improved through this synergistic effect of AuQDs and GC-SPR.
3. In Figure 3, it shows the fluorescence strength of G-AuQDs is enhanced after blending with PEDOT:PSS. It would be better if the authors can explain in detail about its work mechanism.
4. In part 3.2, the authors mentioned that all the fabricated devices are annealed at 150 °C for 45 min. The purpose needs to be presented.

Reviewer: 2

Comments to the Author(s)

Kuntamung et al. used gold quantum dots and grating to enhance the light-harvesting of organic solar cells. By incorporating the down-conversion property of gold quantum dots and surface plasmon resonance of gratings, the authors obtained a 16% increase in power conversion efficiency. The obtained simulation results have a certain significance in light manipulation in photovoltaic devices. This article can be considered publishing on Royal Society Open Science after addressing the following questions:

1. The authors ascribed the device performance enhancement to “AuQDs acted as effective photosensitizers for absorbing light in the UV region and the photoactive layer harvested the emitted visible light”. Why the introduction of gold quantum dots results in the IPCE enhancement in the whole spectrum?
2. The down-conversion efficiency of gold quantum dots is unclear. How many ultraviolet photons absorbed by gold quantum dots can be converted to emitted photons?
3. Why the gold quantum dots boost the hole transporting in PEDOT:PSS rather than trapping the carriers?
4. According to Figure 4, the reflectivity of flat device is below 0.1 around 520 nm, while the G-AuQDs/DVD-R grating-based device has a value around 0.15 that indicating poorer light harvesting. Why the increased reflectivity leads to the increase in IPCE?

Author's Response to Decision Letter for (RSOS-210022.R0)

See Appendix A.

RSOS-210022.R1 (Revision)

Review form: Reviewer 1

Is the manuscript scientifically sound in its present form?

Yes

Are the interpretations and conclusions justified by the results?

Yes

Is the language acceptable?

Yes

Do you have any ethical concerns with this paper?

No

Have you any concerns about statistical analyses in this paper?

No

Recommendation?

Accept as is

Comments to the Author(s)

Interesting work! Hope it can be published successfully!

Review form: Reviewer 2

Is the manuscript scientifically sound in its present form?

Yes

Are the interpretations and conclusions justified by the results?

Yes

Is the language acceptable?

Yes

Do you have any ethical concerns with this paper?

No

Have you any concerns about statistical analyses in this paper?

No

Recommendation?

Accept as is

Comments to the Author(s)

The influences of AuQDs on optical/electrical properties of devices and corresponding mechanisms have been clarified in the revised manuscript. I think the current manuscript is acceptable.

Decision letter (RSOS-210022.R1)

Dear Professor Baba:

Title: The Effect of Gold Quantum Dots/Grating-coupled Surface Plasmons in Inverted Organic Solar Cells

Manuscript ID: RSOS-210022.R1

It is a pleasure to accept your manuscript in its current form for publication in Royal Society Open Science. The chemistry content of Royal Society Open Science is published in collaboration with the Royal Society of Chemistry.

On behalf of the Subject Editor Professor Anthony Stace and the Associate Editor Professor Chaohua Cui.

RSC Associate Editor:
Comments to the Author:
(There are no comments.)

RSC Subject Editor:
Comments to the Author:
(There are no comments.)

Reviewer(s)' Comments to Author:
Reviewer: 1

Comments to the Author(s)
Interesting work! Hope it can be published successfully!

Reviewer: 2

Comments to the Author(s)
The influences of AuQDs on optical/electrical properties of devices and corresponding mechanisms have been clarified in the revised manuscript. I think the current manuscript is acceptable.

Dear Editor,

I am pleased to resubmit a revised and improved manuscript (#RSOS-210022) to *Royal Society Open Science* entitled:

**The Effect of Gold Quantum Dots/Grating-coupled Surface Plasmons
in Inverted Organic Solar Cells**

Kulrisa Kuntamung^{1,2}, Patrawadee Yaiwong^{1,2}, Chutiparn Lertvachirapaiboon¹, Ryousuke Ishikawa¹,
Kazunari Shinbo¹, Keizo Kato¹, Kontad Ounnunkad^{2,3*}, Akira Baba^{1*}

¹*Graduate School of Science and Technology, Niigata University, 8050 Ikarashi-2-nocho, Nishi-ku, Niigata 950-2181, Japan*

²*Department of Chemistry, Faculty of Science, Chiang Mai University, Chiang Mai 50200, Thailand*

³*Center of Excellence for Innovation in Chemistry, Faculty of Science, Research Center on Chemistry for Development of Health Promoting Products from Northern Resources, and Center of Excellence in Materials Science and Technology, Chiang Mai University, Chiang Mai 50200, Thailand*

We have revised the manuscript in accordance with the comments and suggestions by reviewers. We have addressed these comments on a point-by-point basis. A revised version is enclosed for review.

We hope the revised version is now acceptable for publication in *Royal Society Open Science* and we are looking forward to hearing from you in due course.

Sincerely yours,

Akira Baba

Akira Baba, Ph. D.
Professor
Graduate School of Science and Technology
and Faculty of Engineering
Niigata University
8050 Ikarashi 2-nocho, Nishi-ku, Niigata
950-2181 JAPAN
Tel/Fax: +81-25-262-7369
E-mail: ababa@eng.niigata-u.ac.jp

We appreciate the comments of the reviewers and hereby submit a revised and improved manuscript. We have addressed these comments and have incorporated the necessary changes on the manuscript. All the major revisions on the text are shown in RED.

Reviewers' Comments to Author:

Reviewer: 1

Comments to the Author(s)

In this manuscript, the authors have loaded the gold quantum dots (AuQDs) into PEDOT:PSS solution to study the effect on inverted organic solar cells (OSCs), which is assisted with the synergistic effect of grating-coupled surface plasmon resonance (GC-SPR). The authors have provided another possible method to enhance the light-harvesting properties for the OSCs. The results presented in this manuscript are somehow interesting, and it can be published in Royal Society Open Science after some revisions.

1. The conventional device structure is more widely used nowadays for its simple fabrication process and higher performance. However, in this manuscript, the authors only mentioned the effect on the inverted OSCs. To show the universality of this light-harvesting enhancement approach, I would recommend to show some data about the effect on OSCs with conventional structure.

Reply: The conventional and inverted OSCs are different systems in term of the light illumination pathway and device structure. The inverted OSCs demonstrate with the light passing the semiconducting oxide before active layer. It is hardly to compare the performances of two devices. However, we have experience about fabrication of the conventional devices [Refs 18,20,24,29,31 in a manuscript]. In our previous studies, we have demonstrated that introducing plasmonic, diffraction, and scattering structures in conventional OSCs can increase their device efficiency [Refs 15,17,18,20 in a manuscript], which already mentioned and discussed in a manuscript both introduction and result parts (see section 4.1 and 4.3).

In the case of conventional structure, AuQDs cannot be located within the grating-coupled surface plasmon evanescent field [Ref. 31 in a manuscript]. Therefore, in this study, we would like to explore the effect of AuQDs which is located in grating-coupled surface plasmon evanescent field using an inverted OSCs. This explanation is included in an introduction part in a manuscript.

2. Again, to verify its universality, it's better to add at least one more material system, whose photovoltaic performance is improved through this synergistic effect of AuQDs and GC-SPR.

Reply: Recently, we have constructed conventional OSCs by an introduction of AuQDs and plasmonic grating structures [Ref 29 and 31 in a manuscript] into the conventional OSCs system. In these cases, we explored the synergistic effect of AuQDs and gold nanoparticles and also AuQDs and GC-SPR. As mentioned above, in the case of conventional OSCs structure, the AuQDs cannot be located within the strongly enhanced GC-SPR evanescent field. Therefore, we attempt to develop inverted devices using materials and structures mentioned above. As shown in Table 2, the efficiency is improved by this effect. The discussion about the effect of AuQDs and GC-SPR in comparison between conventional OSC structure and inverted OSC structure is included in section 4.1 and 4.3

3. In Figure 3, it shows the fluorescence strength of G-AuQDs is enhanced after blending with PEDOT:PSS. It would be better if the authors can explain in detail about its work mechanism.

Reply: G-AuQDs exhibit strong fluorescence intensity (510 nm) while PEDOT:PSS can absorb a wide range of light in visible and near-infrared (NIR) region, as shown in the figure below. After G-AuQDs are embedded into the PEDOT:PSS matrix, the fluorescence intensity decreases, as compared with that of only G-AuQDs at the same concentration. This is the result of masking and fluorescence-absorption by PEDOT:PSS (this explanation is mentioned in section 4.1).

UV-vis absorption of PEDOT:PSS aqueous solution.

4. In part 3.2, the authors mentioned that all the fabricated devices are annealed at 150 °C for 45 min. The purpose needs to be presented.

Reply: The devices are annealed at 150 °C for 45 min to get the better contact between each film. The procedure is from our previous studies [1-2]. In accordance with the reviewer comment, we have added the information in a manuscript.

References:

[1] Nootchanat S, Pangdam A, Ishikawa R, Wongravee K, Shinbo K, Kato K, Kaneko F, Ekgasit S, Baba A. 2017 Grating-coupled surface plasmon resonance enhanced organic photovoltaic devices induced by Blu-ray disc recordable and Blu-ray disc grating structures. *Nanoscale* 9, 4963-4971. (doi:10.1039/C6NR09951C)

[2] Phetsang S, Phengdaam A, Lertvachirapaiboon C, Ishikawa R, Shinbo K, Kato K, Mungkornasawakul P, Ounnunkad K, Baba A. 2019 Investigation of a gold quantum dot/plasmonic gold nanoparticle system for improvement of organic solar cells. *Nanoscale Adv.* 1, 792-798. (doi:10.1039/C8NA00119G)

Reviewer: 2

Comments to the Author(s)

Kuntamung et al. used gold quantum dots and grating to enhance the light-harvesting of organic solar cells. By incorporating the down-conversion property of gold quantum dots and surface plasmon resonance of gratings, the authors obtained a 16% increase in power conversion efficiency. The obtained simulation results have a certain significance in light manipulation in photovoltaic devices. This article can be considered publishing on Royal Society Open Science after addressing the following questions:

1. The authors ascribed the device performance enhancement to “AuQDs acted as effective photosensitizers for absorbing light in the UV region and the photoactive layer harvested the emitted visible light”. Why the introduction of gold quantum dots results in the IPCE enhancement in the whole spectrum?

Reply: In our previous study, we have demonstrated that a small degree of AuQDs aggregation can create plasmonic-like effect inside the OSCs. This effect is confirmed by FDTD simulation [Ref 24 in a manuscript]. Consequently, the introduction of AuQDs in device are improved by the fluorescence of AuQDs below 450 nm and the plasmonic-like effect due to aggregated AuQDs at longer wavelength region, resulting in the IPCE enhancement in the whole spectrum. We have added the reference information in the section 4.3.

2. The down-conversion efficiency of gold quantum dots is unclear. How many ultraviolet photons absorbed by gold quantum dots can be converted to emitted photons?

Reply: The structure of device has multilayer consisting of ITO/TiO₂/P3HT:PCBM/PEDOT:PSS:AuQD/Ag, in which each layer can absorb the ultraviolet photons. Hence, it is hardly to calculate ultraviolet photons absorbed by AuQDs embedded in PEDOT:PSS matrix since the incident ultraviolet photons actually pass the two film layers before getting into the PEDOT:PSS layer. Moreover, the PEDOT:PSS would adsorb the incident photons before it can excite AuQDs. In addition, when AuQDs are in PEDOT:PSS, fluorescence intensity is lowered, as shown in Fig 3. The emission of fluorescence from AuQDs is also hardly determined due to masking by PEDOT:PSS. The harvestable fluorescence to the active layer is not 100% of initial emitted light from AuQD. However, we have considered that the effect of fluorescence emission from AuQDs within device can be determined by the power conversion efficiency (PCE) as compared to that of reference inverted OSC, as shown in Table 1. This suggests that the fluorescence phenomenon of AuQDs results in a significant PCE increment.

3. Why the gold quantum dots boost the hole transporting in PEDOT:PSS rather than trapping the carriers?

Reply: In the impedance measurements, R_{ct} values mostly corresponds to the bulk resistance of the TiO₂/P3HT:PCBM and R_s values corresponds to the resistance of PEDOT:PSS in our model. As pointed out by the reviewer, the R_s value was slightly lowered by adding the AuQDs, indicating the increased number of the charge carrier in PEDOT:PSS. The reason for this might be due to the enhanced hole injection at the PEDOT:PSS/P3HT:PCBM interface. In addition to the photocarrier generation via fluorescence of AuQDs, excited electrons within AuQDs at the interface of PEDOT:PSS-AuQDs/P3HT:PCBM might be injected into the conduction band of the adjacent PCBM and, at the same time, the holes might be injected into the valence band of PEDOT:PSS, which could also contribute to the increased short-circuit current. We have added the explanation in section 4.5.

4. According to Figure 4, the reflectivity of flat device is below 0.1 around 520 nm, while the G-AuQDs/DVD-R grating-based device has a value around 0.15 that indicating poorer light harvesting. Why the increased reflectivity leads to the increase in IPCE?

Reply: The lowered reflectivity in the flat device was observed at higher incident angle. This might be due to multiple reflection in the device. However, the photovoltaic measurement was carried out at vertical light incidence. Although we could not measure the reflectivity at

vertical light incidence (0 deg.) due to the measurement configuration limit, the reflectivity at lower incident angle in the figure is not lower than that of grating device (about same or lower reflectivity at 20 deg.). An important point here is that the device with grating harvests the light with broad wavelength region due to the plasmonic effect. We have added short explanation in section 4.2